# Phloridzin Docosahexaenoate Inhibits Spheroid Formation by Breast Cancer Stem Cells and Exhibits Cytotoxic Effects against Paclitaxel-Resistant Triple Negative Breast Cancer Cells

**DOI:** 10.3390/ijms241914577

**Published:** 2023-09-26

**Authors:** Wasundara Fernando, Rikki F. Clark, H. P. Vasantha Rupasinghe, David W. Hoskin, Melanie R. Power Coombs

**Affiliations:** 1Department of Pathology, Dalhousie University, Halifax, NS B3H 4H7, Canada; 2Biology Department, Acadia University, Wolfville, NS B4P 2R6, Canada; 3Department of Plant, Food, and Environmental Sciences, Dalhousie University, Truro, NS B2N 5E3, Canada

**Keywords:** breast cancer, cancer stem cells, phytochemical, phloridzin docosahexaenoate

## Abstract

The eradication of cancer stem cells (CSCs) is vital to successful cancer treatment and overall disease-free survival. CSCs are a sub-population of cells within a tumor that are defined by their capacity for continuous self-renewal and recapitulation of new tumors, demonstrated in vitro through spheroid formation. Flavonoids are a group of phytochemicals with potent anti-oxidant and anti-cancer properties. This paper explores the impact of the flavonoid precursor phloridzin (PZ) linked to the ω-3 fatty acid docosahexaenoate (DHA) on the growth of MCF-7 and paclitaxel-resistant MDA-MB-231-TXL breast cancer cell lines. Spheroid formation assays, acid phosphatase assays, and Western blotting were performed using MCF-7 cells, and the cell viability assays, Annexin-V-488/propidium iodide (PI) staining, and 7-aminoactinomycin D (7-AAD) assays were performed using MDA-MB-231-TXL cells. PZ-DHA significantly reduced spheroid formation, as well as the metabolic activity of MCF-7 breast cancer cells in vitro. Treatment with PZ-DHA also suppressed the metabolic activity of MDA-MB-231-TXL cells and led to apoptosis. PZ-DHA did not have an observable effect on the expression of the drug efflux transporters ATP-binding cassette super-family G member 2 (ABCG2) and multidrug resistance-associated protein 1 (MRP1). PZ-DHA is a potential treatment avenue for chemo-resistant breast cancer and a possible novel CSC therapy. Future pre-clinical studies should explore PZ-DHA as a chemo-preventative agent.

## 1. Introduction

Cancer stem cells (CSCs) are a sub-population of cells within a tumor that can continuously self-renew and recapitulate a heterogenous tumor through differentiation [1,2]. CSCs are generally considered rare, but their prevalence may depend upon the tumor type [1]. A number of stemness-associated proteins, such as cluster of differentiation 133 (CD133), CD44, CD24, [3,4] and aldehyde dehydrogenase (ALDH) enzyme activity [5] are used to identify CSCs. The correspondence between stemness markers and high patient mortality highlights the clinical relevance of CSCs, which are more tumorigenic and drug-resistant when compared to other cancer cells from the same tumor. Having a CSC-targeted therapy is important for killing chemotherapy-resistant CSC and may be essential for preventing cancer relapse.

Triple negative breast cancers (TNBC) are phenotypically defined as lacking estrogen (ER) and progesterone (PR) receptors, as well as human epidermal growth factor receptor 2 (HER2) [6]. TNBC is an aggressive type of breast cancer with poor prognosis [7]. This leaves fewer treatment options, as a number of systemic therapies target the receptors that TNBCs lack. Novel strategies are needed to target TNBC, especially in the case of metastatic disease. Chemotherapy is the major therapeutic option to manage TNBC and due to the lack of targeted therapies for TNBC, this disease is associated with the development of drug resistance and, therefore, recurrence with metastasis. The selectivity of most chemotherapeutic drugs is poor and causes considerable toxicity to normal cells. Therefore, it is important to consider selectivity and toxicity carefully when developing potential new therapies to treat TNBC. A frequent cause of drug resistance in cancer cells is the upregulation of drug efflux transporters ATP-binding cassette super-family G member 2 (ABCG2) and multidrug resistance-associated protein 1 (MRP1).

Plant-based medicines play a significant role in the prevention and treatment of several human diseases, including cancer. Phytochemicals, which are naturally occurring compounds found in plants, are key resources in cancer drug discovery [8]. Studies have shown that regular intake of phytochemicals such as flavonoids is inversely proportional to breast cancer risk in humans [9,10,11]. Furthermore, many studies suggest that anti-oxidant activity of dietary flavonoids plays a role in alleviating adverse side effects of chemotherapy, as well as in cancer chemo-prevention [12,13,14]. Flavonoid-induced cytotoxic and chemo-preventive activities are linked to the ability of flavonoids to impact biological systems such as inhibition of growth-promoting protein kinases, cell cycle progression, metastasis, angiogenesis, multi-drug resistance, metabolism of carcinogens, and pro-oxidant enzymes [15]. Flavonoids possess a wide range of anti-cancer properties. The anti-oxidant effects of flavonoids offer protection from reactive oxygen species [16,17], which may damage cells and lead to carcinogenic changes. Phloridzin (PZ) is a flavonoid precursor found in apple peels, seeds, and apple juice, as well as the bark, roots, shoots, and leaves of the apple tree [18]. Like other flavonoids, PZ is often characterized by limited cellular uptake, and poor bioavailability. Therefore, attempts have been made to increase its efficacy through the attachment of lipophilic compounds. Fatty acid esters of PZ are significantly more effective in fighting tumors than either parent compound alone. Docosahexaenoate (DHA) is an ω-3 fatty acid found in fish oil [19]. PZ-DHA is a fatty acid ester of PZ (Figure 1) [20] formed through an acylation reaction. Of the six fatty acid esters tested, PZ-DHA was the most effective chemotherapeutic agent [21].

Treatment with PZ-DHA significantly downregulates the expression of cancer-promoting genes in HepG2 cells, while having little cytotoxic effect on normal human and rat hepatocytes [21]. Furthermore, PZ-DHA inhibits the growth of three forms of leukemia cells, namely THP1, K562, and Jurkat, in vitro [20,22]. PZ-DHA also has a selective cytotoxic effect on breast cancer cells in vitro and in vivo (MDA-MB-231, MDA-MB-468, 4T1, MCF-7, and T-47D), as well as the capacity to inhibit metastasis of TNBC cells MDA-MB-231 and 4T1 at sub-cytotoxic levels in non-obese diabetic severe combined immunodeficient mice and immunocompetent BALB/c mice, respectively [19,20]. A variety of mechanisms have been proposed to explain the anti-cancer effects of PZ-DHA and its parent components. One study demonstrated that PZ-DHA exhibits anti-inflammatory effects on THP-1-derived macrophages through inhibition of Nuclear Factor (NF)-κB [23]. Our publication analyzing the effect of PZ-DHA on TNBC cells noted a decline in GTPase proteins associated with cell migration, proteins involved in the epithelial-mesenchymal transition, and Akt/PI3K and mitogen-activated protein kinase (MAPK) signaling [20]. Likewise, DHA alone and another ω-3 fatty acid inhibit proliferation and induce apoptosis in MDA-MB-231 human breast cancer cells in vitro, potentially through inhibition of the Akt/NF-κB signal transduction pathway [24]. It is unclear whether acylation with PZ interrupts these particular anti-cancer mechanisms of DHA. However, our previous studies have shown that PZ-DHA possesses greater anti-cancer [19,20,22] and anti-inflammatory [23] potential than either of its parent compounds.

Given the anti-cancer and anti-inflammatory potential of PZ-DHA, it is hypothesized that this compound will inhibit stem cell activity in MCF-7 breast cancer cells, inhibit metastatic activity in MDA-MB-231 and MDA-MB-231-TXL TNBC cells, and prove cytotoxic to all three cell types. It is also hypothesized that PZ-DHA will exhibit greater inhibitory and cytotoxic effects against TNBC cells than either PZ or DHA alone. PZ-DHA has the potential to address an area of therapeutic need and importance and this paper will further the understanding of PZ-DHA in the treatment of chemo-resistant TNBC.

## 2. Results

### 2.1. PZ-DHA Suppresses Primary and Secondary Spheroid Formation by Breast Cancer Cells

PZ-DHA reduced spheroid formation by MCF-7 cells and led to a decline in metabolic activity in MCF-7 spheroids, as evidenced by decreased acid phosphatase activity in the presence of a sub-cytotoxic concentration (50 μM) (*p* < 0.001) (Figure 2A,B). Moreover, PZ-DHA suppressed the establishment of secondary spheroids by MCF-7 cells isolated from primary spheroids (Figure 2C).

### 2.2. PZ-DHA Attenuates the Growth of Paclitaxel-Resistant MDA-MB-231 (MDA-MB-231-TXL) Cells

PZ-DHA changed the morphology of MDA-MB-231-TXL cells and reduced their prevalence in the culture, indicating cytotoxic activity and/or a decline in cell proliferation (Figure 3A). 3-(4,5-dimethylthiazol-2-yl)-2,5-diphenyl-2H-tetrazolium bromide (MTT) assays were used to evaluate the metabolic activity of cells following treatment with PZ-DHA or its parent compounds for 24 h. Low concentrations of PZ-DHA (10 and 25 μM) did not cause considerable changes in cell metabolism; however, longer treatment led to dose-dependent and statistically significant reductions in metabolic activity. Concentrations of 50 and 100 μM PZ-DHA suppressed metabolic activity considerably after 24 h, and further after 48 h (*p* < 0.001) (Figure 3B,C). Paclitaxel-resistant TNBC cells (MDA-MB-231-TXL) were less sensitive to treatment with PZ-DHA compared with their non-paclitaxel-resistant counterparts (MDA-MB-231) after 24 h after culture (*p* < 0.05). This disparity was less evident 48 h after treatment (Figure 3D).

Flow cytometry of MDA-MB-231-TXL cells treated with PZ-DHA revealed a notable increase in cell death (Figure 4A). Analysis by flow cytometry using Annexin V/propidium iodide staining indicated significant amounts of early and late apoptosis/necrosis among the cells (Figure 4B). Treatment with PZ-DHA led to a reduction in the pro-caspase 3 level, without an increase in cleaved-caspase 3. Treatment with DHA resulted in a marked increase in cleaved-caspase 3 (fraction 2, MW = 17 kDa), indicating that caspase-3 activation may be involved in DHA’s anti-cancer mechanism of action (Figure 4C). Western blot analysis did not reveal any significant effects of PZ-DHA, nor its parent compounds, on the expression of drug-efflux transporters ABCG2 (MW = 65 kDa) and MRP1 (MW = 190 kDa), encoded by ATP Binding Cassette Subfamily C Member 1 (ABCC1) (Figure 4D).

## 3. Discussion

There is an urgent need for the development of therapeutic strategies for TNBC due to the poor prognosis and lack of targeted therapies for this aggressive cancer [8]. Numerous flavonoids possess anti-proliferative, anti-metastatic, and cytotoxic properties against CSCs in vitro [26,27,28], and many studies have examined the effects of flavonoids as adjuvants to conventional cancer therapy [29,30,31]. Flavonoids are sensitive to environmental factors, making them chemically unstable, and are rapidly metabolized in the body, which results in low bioavailability and reduces their therapeutic potential [8]. Modified flavonoids like PZ-DHA have the advantage of being a promising compound for the prevention and treatment of TNBC due to improved stability and bioavailability. Previous studies have demonstrated the cytotoxicity of PZ-DHA and other flavonoids derived from apple peels, such as apple flavonoid-enriched fraction (AF4), against MDA-MB-231 TNBC cells [19,20,32], and the results of this study are consistent with these previous findings. However, the present study is the first to examine the effects of PZ-DHA against paclitaxel-resistant TNBC cells. The results of this study show that PZ-DHA inhibited the metabolic activity of paclitaxel-resistant TNBC cells (MDA-MB-231-TXL) and exerted cytotoxicity in a time- and dose-dependent manner. Given that paclitaxel is a standard pre-adjuvant and adjuvant therapy to improve prognosis in breast cancer patients, paclitaxel resistance is a serious issue that leads to poor prognosis for patients and especially those diagnosed with TNBC, for which there are already fewer treatment options than for other forms of breast cancer [33,34]. The discovery that PZ-DHA exhibits cytotoxic activity against paclitaxel-resistant TNBC cells in vitro is significant because it presents a potential therapy to improve the prognosis of patients for whom paclitaxel and other standard chemotherapies may not be effective. Research into the efficacy of PZ-DHA against paclitaxel-resistant TNBCcells in animal models is warranted, and the use of patient-derived xenograft (PDX) models is recommended. Examining PZ-DHA in a TNBC metastasis model would be beneficial since dormant metastatic cancer cells may have become disseminated through the circulation even before surgery or the start of chemotherapy and/or radiation treatment [35,36,37]. Since PZ-DHA shows potent and safe anti-metastatic activity in vivo, it should be determined whether PZ-DHA inhibits the in vivo metastasis in these contexts using a mouse model of metastatic breast cancer.

CSC, a sub population of cells in tumors, demonstrate distinct phenotypic and genotypic signatures allowing them to escape chemotherapies that targets rapidly proliferating cancer cells [38,39,40]. Suppression of stem cell-like activity in breast cancer cells by PZ-DHA was evident by the PZ-DHA-mediated inhibition of MCF-7 spheroid growth. To our knowledge, this is the first study to establish the impact of PZ-DHA against breast CSC activity in vitro via spheroid formation assays using MCF-7 cells. PZ-DHA reduced phosphatase activity and spheroid size in primary spheroids at a concentration that is sub-cytotoxic to healthy mammary cells. PZ-DHA also suppressed the formation of secondary spheroids derived from primary spheroids. These effects were shown using MCF-7 cells expressing ER. A TNBC cell line such as MDA-MB-231 would have been ideal to investigate the effects of PZ-DHA on TNBC cells. Unfortunately, the inadequate cell–cell interactions of TNBC cells resulted in a lack of MDA-MB-231 cell spheroids with clear margins. CSCs are thought to play a significant role in all stages of cancer, from the initiation of a primary tumor to the establishment of secondary metastases, as well as the development of therapy resistance [41]. PZ-DHA inhibited the metabolic activity of MDA-MB-231-TXL in a time- and concentration-dependent manner. PZ-DHA-induced MDA-MB-231-TXL cell death was also dose dependent. PZ-DHA treatment decreased pro-caspase 3 levels but did not increase cleaved-caspase 3. A previous study has shown DHA-induced caspase 3/7 activation in MDA-MB-231 cells [19]. DHA increased levels of cleaved-caspase 3 while decreasing pro-caspase 3 in MDA-MB-231-TXL cells. PZ, DHA, and PZ-DHA did not have any impact on the basal expression of drug transporters (ABCG2 or MRP1) in MDA-MB-231-TXL cells. The affinity of flavonoids to ABC family drug efflux transporters has been reported previously [42,43]. Therefore, the interaction of PZ-DHA with other drug transporters should be evaluated during future studies.

If PZ-DHA inhibits CSC metabolism in vivo, it may be a useful therapy to assist in preventing metastasis and the development of therapy resistance as PZ-DHA does not increase drug efflux transporters. Use of a 3-dimensional (3D) dynamic cell culture system is recommended to further assess the anti-CSC activity of PZ-DHA because this culture system more closely mirrors the dynamism of the tumor microenvironment [44]. A 3D dynamic cell culture system is thus able to provide a model which more accurately demonstrates the effects of anti-metastatic drugs on CSC function and metastasis in a living biological system than sphere formation assays alone [44]. Most importantly, PZ-DHA activity against CSCs should be examined using animal models, preferably using CSCs originating from PDX, to measure the potential of PZ-DHA to inhibit or eliminate CSCs in vivo. It should be noted that studies using CSCs must identify them functionally rather than phenotypically, as CSC surface markers are expressed heterogeneously among samples and cannot be applied universally to all individuals [45]. CSCs must be validated through their characteristic ability to self-renew and recapitulate a primary tumor [46]. This can be confirmed through flow cytometry, colony formation assays, sphere formation assays, and xenograft assays [47].

## 4. Materials and Methods

### 4.1. Breast Cancer Spheroid Formation Assay

MCF-7 cells (1 × 10^4^ cells) were grown in ultra-low adherent cell culture plates in spheroid growth medium (F12 medium supplemented with 20 ng/mL basic fibroblast growth factor, 20 ng/mL epidermal growth factor, 100 U/mL penicillin, 100 μg/mL streptomycin, and B27 serum-free supplement) for 48 h and treated with PZ, DHA, PZ-DHA (50 μM), vehicle control or medium control alone for 72 h. Following culture, spheroids were photographed using a phase contrast microscope (Nikon eclipse TS 100 phase contrast microscope, Melville, NY, USA) and the viability of cells in spheroids was determined using an acid phosphatase assay [32,48]. Secondary spheroids were formed in the same manner as primary spheroids using cells derived from the primary spheroids.

### 4.2. Acid Phosphatase Assay

The viability of drug-treated MCF-7 cells in breast cancer spheroids was measured using the acid phosphatase assay. This assay was chosen over an MTS assay because the acid phosphatase assay buffer lyses the cells prior to analysis, allowing the determination of the total phosphatase activity of cells in the entire spheroid population. For acid phosphatase assays, spheroids were washed using 1 × PBS and resuspended in 1 mL acid phosphatase assay solution (0.1 M sodium acetate at pH 5.5, 0.1% Triton-X-100, 4 mg/mL phosphatase substrate) and incubated for 2 h at 37 °C in the dark. The reaction was stopped by adding 25 μL 1 N NaOH. Absorbance was measured at 405 nm and % acid phosphatase activity was determined as in the formula:% Relative acid phosphatase activity = (*A*_*T*_ − *A*_*B*_⁄*A*_*C*_ − *A*_*B*_) × 100(1)
where, *A*_*T*_: absorbance of cells treated with drugs; *A*_*C*_: absorbance of cells treated with vehicle control; *A*_*B*_: absorbance of the blank.

### 4.3. MTT Assay

The inhibitory effect of PZ-DHA and its parent compounds on the growth in MDA-MB- 231-TXL cells (in comparison to parent MDA-MB-231) was tested using MTT assays [48]. Cells were seeded in flat-bottom 96-well plates. MDA-MB-231 and paclitaxel-resistant MDA-MB-231 (MDA-MB-231-TXL) cells were treated with PZ, DHA, and PZ-DHA at 10–100 μM. At the end of the treatment, 10 μL of MTT reagent was added (454 μg/mL) and incubated at 37 °C for 3 h. Formazan crystals were collected by centrifuging plates at 1400× *g* for 5 min. The supernatant was removed, and formazan crystals were dissolved in 100 μL of DMSO. The absorbance was measured at 570 nm using an Expert microplate reader (Admiral Place, Guelph, ON, USA) and the % metabolic activity was determined using the formula:% Relative metabolic activity = (*A*_*T*_⁄*A_C_*) × 100.(2)

### 4.4. Annexin-V-488/PI Staining

Induction of early apoptosis and late apoptosis/necrosis by PZ-DHA and its parent compounds was determined using Annexin-V-488/propidium iodide (PI) staining [48]. MDA-MB-231-TXL cells grown in monolayers were seeded at a density of 1 × 10^4^ (24 h) cells per well in 6-well plates and cultured overnight at 37 °C to induce cell adhesion. Adherent cells were treated with 50 and 100 μM of PZ, DHA, PZ-DHA, vehicle control, or medium control alone at 37 °C for 24 h. Cells grown in monolayers were harvested using TrypLE express and combined with their respective medium. Cells were centrifuged and rinsed with 1 × PBS and incubated with Annexin-V-488 (1 μg/mL) and PI (1 μg/mL) in staining buffer (10 mM HEPES, 10 mM NaCl, and 5 mM CaCl_2_) at room temperature for 15 min. Flow cytometric analysis was performed using a FACSCalibur instrument (BD Bioscience, Mississauga, ON, USA). Cells (1 × 10^4^) were counted per sample, and both live and dead cells were included in counts.

### 4.5. 7-AAD Assay

7-AAD distinguishes dead cells from viable cells by intercalating between cytosine and guanine bases of DNA of dead cells. MDA-MB-231-TXL were plated at a density of 5 × 10^4^ cells per well in 6-well plates. The cells were cultured overnight to induce cell adhesion. Adherent cells were treated with 25, 50, and 100 μM of PZ, DHA, PZ-DHA, vehicle control or medium control alone and cultured for 48 h. Cells were harvested using TrypLE Express, washed, and resuspended in 1 × PBS. Cells in PBS were incubated with 5 μL of 7-AAD viability staining solution (eBioscience Inc. San Diego, CA, USA) at room temperature for 5 min. Flow cytometric analysis was performed using a FACSCalibur instrument (BD Bioscience, Mississauga, ON, USA). Cells (1 × 10^4^ cells/sample) were counted; both live and dead cells were included in counts.

### 4.6. Western Blotting

The effect of PZ-DHA on drug efflux transporters was investigated using Western blot analysis. Protein-rich cell lysates (10–40 μg proteins) were loaded into 7.5, 10, 12, or 15% SDS polyacrylamide gels, as appropriate for the protein being studied. Proteins were transferred to nitrocellulose membranes, and blots were incubated in 5% non-fat milk or 5% BSA prepared in Tween-tris buffered saline (TBS) [0.25 M Tris (pH 7.5), 150 mM NaCl and 0.2% Tween-20] for 1 h to block nonspecific binding. PZ-DHA-mediated expression of drug efflux transporters were tested by probing the blots with Ab against ABCG2 (anti-ABCG2 rabbit monoclonal Ab (#4477) purchased from Cell Signaling Technology (Danvers, MA, USA)) and MRP1 (anti-MRP1 mouse monoclonal Ab (SC-18835) purchased from Santa Cruz Biotechnology (Santa Cruz, CA, USA)). Apoptosis induction was tested by probing the blots with antibodies (anti-caspase 3 rabbit monoclonal Ab (#9662) purchased from Cell Signaling Technology (Danvers, MA, USA)) for pro-caspase 3 and cleaved caspase 3. Then, the blots were washed thoroughly with 0.05% Tween-TBS and probed with HRP-conjugated donkey anti-rabbit or goat anti-mouse IgG Ab (HRP-conjugated donkey anti-rabbit (#7074) and goat anti-mouse IgG Ab were purchased from Cell Signaling Technology Inc. (Danvers, MA, USA)) by incubating for 1 h at room temperature. Even protein loading was confirmed by probing the blots with HRP-conjugated rabbit anti-actin Ab (HRP-conjugated rabbit anti-β-actin (#12620) purchased from Cell Signaling Technology Inc. (Danvers, MA, USA)) followed with HRP-conjugated donkey anti-rabbit IgG Ab. The proteins of interest were visualized by X-ray film exposure or ChemidocTouchTM imaging system (Bio-Rad Laboratories, Mississauga, ON, USA).

## 5. Conclusions

Numerous flavonoids exhibit potent anti-cancer effects in vitro through a variety of mechanisms and these effects span many cancer types, as has been demonstrated with a variety of assays [26,27,28]. The bioavailability of flavonoids can be augmented by chemically attaching other compounds, as has been performed in the case of PZ-DHA. To the best of our knowledge, this study is the first to demonstrate that the conjugated phytochemical PZ-DHA possesses inhibitory activity against breast CSCs, as well as time- and dose-dependent cytotoxic effects against paclitaxel-resistant TNBC cells. PZ-DHA has strong potential as a novel chemotherapy or as an adjuvant to conventional cancer treatments. Given the high risk of relapse associated with CSCs and the difficulty in treating TNBC, the ability of PZ-DHA to inhibit CSC activity and trigger apoptosis in TNBC cells makes it an especially promising candidate drug to improve the prognoses of patients with difficult-to-treat cancers such as TNBC. Further investigation of PZ-DHA as an anti-cancer agent is warranted and its potential as a chemo-preventative agent should be explored.

## Figures and Tables

**Figure 1 ijms-24-14577-f001:**
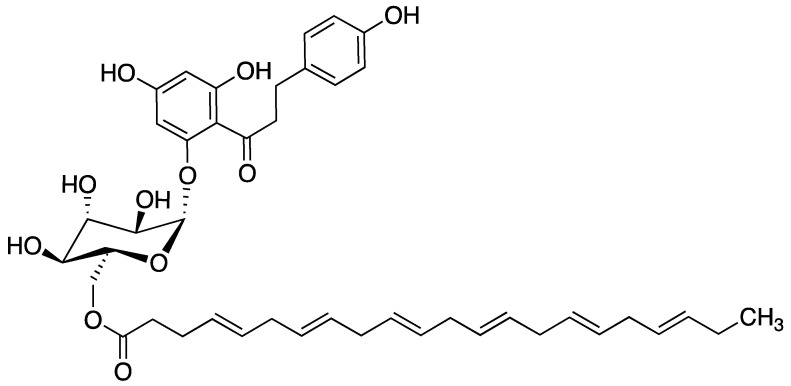
The chemical structure of phloridzin-docosahexaenoate (C_43_H_54_O_11_, PZ-DHA), a fatty acid ester of the glycosylated flavonoid phloridzin (PZ) [20]. Illustration prepared using ChemDraw Prime 16.0.

**Figure 2 ijms-24-14577-f002:**
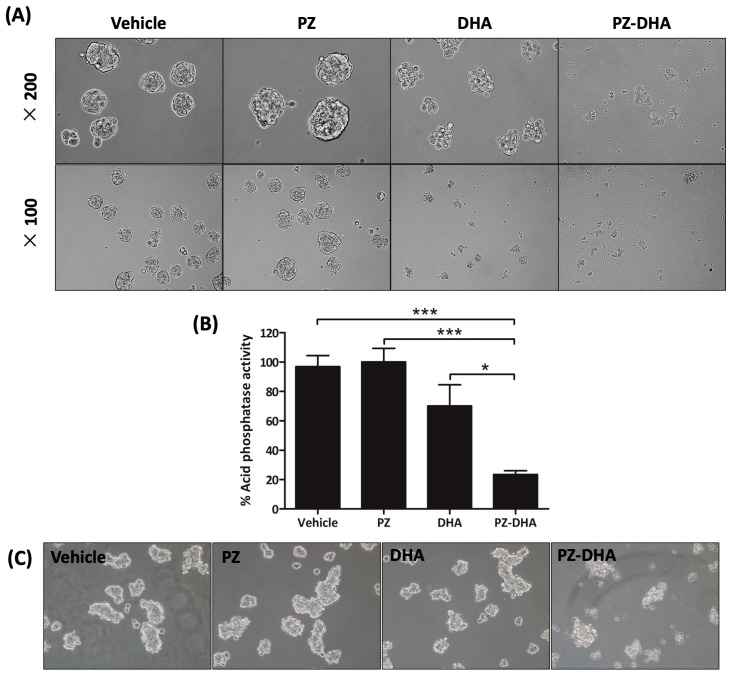
PZ-DHA suppresses in vitro spheroid formation by breast cancer cells. Spheroids were grown from ER+ breast cancer cells (MCF-7) using ultra-low adherent cell culture plates in spheroid growth medium. After 48 h, the cells were treated with PZ, DHA, PZ-DHA (50 μM), or vehicle (DMSO) and cultured for another 72 h. (**A**) Treated spheroids were photographed at 100× and 200× using a phase contrast microscope. (**B**) An acid phosphatase assay was used to determine metabolic activity through the detection of the named enzyme produced by living cells. Data are mean acid phosphatase activity ± SEM of three independent experiments. ANOVA multiple means comparison statistical method was completed and differences between means were compared with Tukey’s test; * *p* < 0.05, and *** *p* < 0.001. (**C**) Spheroids were broken up into single-cell suspensions and treated with PZ, DHA, or PZ-DHA. The growth in secondary spheroids was observed and photographed at 100× using a phase contrast microscope after 72 h.

**Figure 3 ijms-24-14577-f003:**
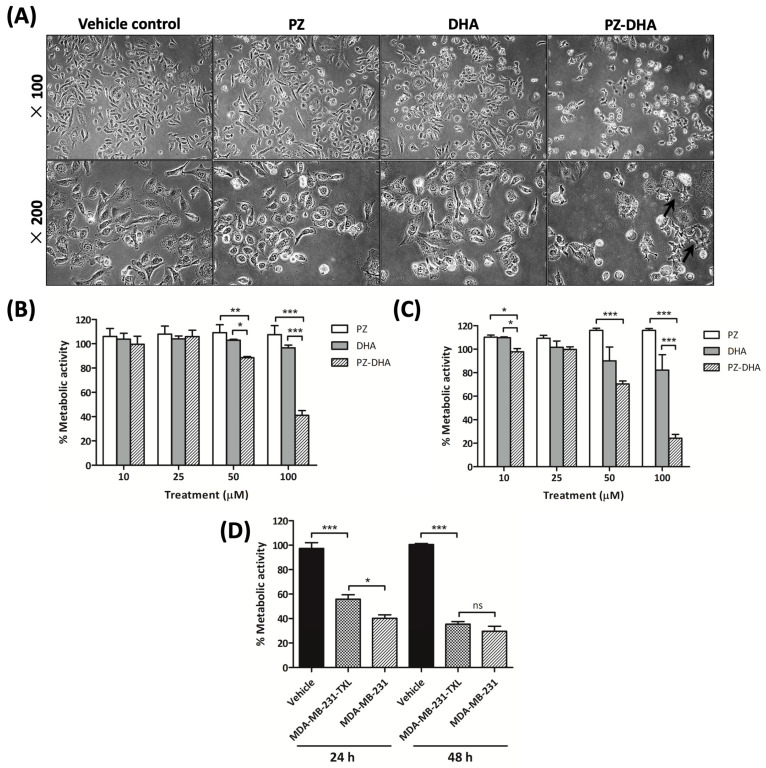
PZ-DHA suppresses the metabolic activity of paclitaxel-resistant MDA-MB-231 (MDA-MB-231-TXL) TNBC cells in a dose- and time-dependent manner. Paclitaxel-resistant TNBC cells (MDA-MD-231-TXL) and MDA-MB-231 cells were treated with PZ, DHA, PZ-DHA (10–100 μM), vehicle (DMSO), or medium alone and cultured for 24 or 48 h. (**A**) Cells were visualized at 100× and 200× after 48 h to determine morphology. An MTT assay was performed to determine metabolic activity following (**B**) 24 and (**C**) 48 h of treatment. MTT reagent (tetrazolium dye) is reduced by cellular oxidoreductases in living cells to form insoluble formazan, which is detected through a plate reader. (**D**) The effects of 50µM PZ-DHA on the metabolic activity of MDA-MB-231 and MDA-MD-231-TXL cells, as well as cell-free vehicle were compared. Data are mean metabolic activity ± SEM from three independent experiments. ANOVA multiple means comparison statistical method was completed and differences between means were compared with Tukey’s test; ns: not significant, * *p* < 0.05, ** *p* < 0.01, and *** *p* < 0.001.

**Figure 4 ijms-24-14577-f004:**
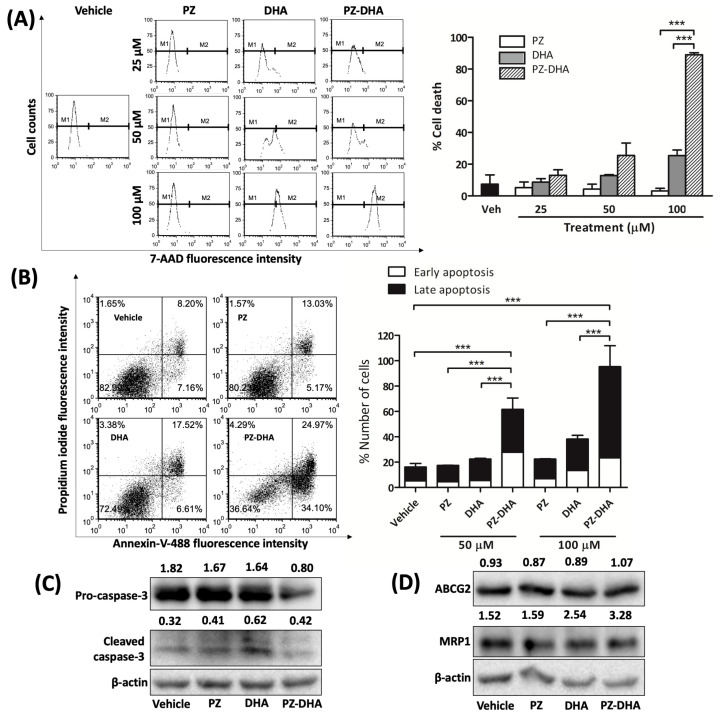
PZ-DHA is cytotoxic to paclitaxel-resistant TNBC cells (MDA-MB-231-TXL). (**A**) Paclitaxel-resistant TNBC cells (MDA-MB-231-TXL) were treated with sub-cytotoxic concentrations of PZ, DHA, PZ-DHA (25, 50 or 100 μM) or vehicle (DMSO), cultured for 48 h, and stained with 7-AAD for flow cytometric analysis. Cell viability is determined as 7-amino-actinomycin D (7-AAD) is excluded by living cells, binding only to the DNA of dead cells [25]. Within the representative histograms, M1 indicates live cells and M2 indicates dead cells. Data are shown as mean % cell death ± SEM of three independent replications. (**B**) Annexin-V-488/PI and flow cytometry were used to detect PI activity following treatment of MDA-MB-231-TXL cells with PZ, DHA, and PZ-DHA (50 or 100 μM) or vehicle (DMSO) for 24 h as an indicator of early and late apoptosis. Representative dot plots include data from solely the 50 μM treatment group and vehicle. Bar graph data indicate the mean % of cells in early apoptosis and late apoptosis/necrosis ± SEM across three independent replications, including both treatment groups and vehicle. ANOVA multiple means comparison statistical method was completed and differences between means were compared with Tukey’s test; *** *p* < 0.001. Western blot analyses using protein-rich cell lysates evaluated the effect of PZ, DHA, and PZ-DHA (20 μM) on (**C**) caspase 3 activation, indicated by pro-caspase-3 and cleaved caspase-3, as well as the expression of (**D**) drug efflux transporters, ABCG2 and MRP1. A β-actin probe served to confirm equal protein loading. Densiometric values are listed above each band and were calculated with image J in comparison to β-actin and normalized to the medium control.

## Data Availability

Further information about the data presented in this study are available on request from the corresponding author.

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
