# Peer review of "Phloridzin Docosahexaenoate Inhibits Spheroid Formation by Breast Cancer Stem Cells and Exhibits Cytotoxic Effects against Paclitaxel-Resistant Triple Negative Breast Cancer Cells"

_ijms, 2023, doi:10.3390/ijms241914577_

Round 1
Reviewer 1 Report
In this research article, the authors have explored the potential of phloridzin docosahexaenoate (PH-DHA) to prevent spheroid formation and growth in MCF-7 and MDA-MB-231-TXL cells respectively. Through microscopy, they have shown the prevention of spheroid formation in MCF-7 cells and demonstrated the decrease in metabolic activity of these cells through measurement of acid phosphatase activity. MTT assays, annexin-V-488/Propidium iodide staining and 7-AAD staining indicated that PZ-DHA treatment induced apoptosis in MDA-MB-231-TXL cells. This compound, however, did not have an effect on the expression of drug efflux transporters ABCG2 and MRP1. The authors conclude by indicating that further investigation of PZ-DHA as an anti-cancer drug is warranted.
Comments:
This is a succinct and well written manuscript that demonstrates the potential of PZ-DHA to serve as an anti-cancer drug, especially against triple negative breast cancers. The experiments are well designed and the results compel the reader to explore the possibility of using PZ-DHA as a chemotherapeutic agent. The quality of the manuscript can be improved if the authors address the following concern
· In Figure 2C, the authors demonstrate that secondary spheroid formation is prevented upon PZ-DHA treatment for 72h. It may be useful to capture and show the spheroids at 0h of treatment to compare it with the 72h treatment to give the readers an idea of the initial spheroid size.
Author Response
Thank you for the suggestion. Formation of mammospheres takes at least 72 hours. Therefore, at 0 hours there will not be mammospheres but only the seeded individual cells.
Reviewer 2 Report
The article, " Phloridzin Docosahexaenoate Inhibits Spheroid Formation by Breast Cancer Stem Cells and Exhibits Cytotoxic Effects Against Paclitaxel-Resistant Triple Negative Breast Cancer Cells" by Wasundara Fernando et al., is informative, but there are some issues that need to be addressed.
1. It is imperative that the authors provide their own justification since the relevance of the study has already been explored in various in vitro and in vivo experiments in PubMed: Carcinogenesis. 2016 Oct;37(10):1004-1013. doi: 10.1093/carcin/bgw087; Cancer Lett. 2019 Nov 28;465:68-81. doi: 10.1016/j.canlet.2019.08.015; Sci Rep. 2020 Dec 7;10(1):21391. doi: 10.1038/s41598-020-78369-0. and hence I do not find any innovative information in the study
2. All experiments were carried out using MCF-7 and MDA-MB-231-TXL. However, an additional control cell line (MCF-10) should be used to interpret and validate the key findings of the study
3. For western blot analysis, authors should include molecular weight and densitometry of bands (Fig 4C, 4D etc).
4. Authors must perform a clonogenic assay to determine the anti-proliferative properties of Phloridzin Docosahexaenoate using breast cancer cells
5. Authors did not perform cell proliferation markers (Ki67, PCNA, BrdU, etc.), and cell cycle markers using the given cell lines to confirm the anti-proliferative properties of Phloridzin Docosahexaenoate.
6. The authors did not perform downstream or upstream pathways to confirm the ability of Phloridzin Docosahexaenoate to proliferate, migrate and invade the tumor.
7. Authors did not perform any in vivo experiments in the present study. They did not get consistent outcomes in the anti-proliferative properties of Phloridzin Docosahexaenoate in tumor-bearing breast cancer animals.
Moderate editing of English language required
Author Response
Reviewer 2 summary and comments/Responses
The article, " Phloridzin Docosahexaenoate Inhibits Spheroid Formation by Breast Cancer Stem Cells and Exhibits Cytotoxic Effects Against Paclitaxel-Resistant Triple Negative Breast Cancer Cells" by Wasundara Fernando et al., is informative, but there are some issues that need to be addressed.
- It is imperative that the authors provide their own justification since the relevance of the study has already been explored in various in vitro and in vivo experiments in PubMed: 2016 Oct;37(10):1004-1013. doi: 10.1093/carcin/bgw087; Cancer Lett. 2019 Nov 28;465:68-81. doi: 10.1016/j.canlet.2019.08.015; Sci Rep. 2020 Dec 7;10(1):21391. doi: 10.1038/s41598-020-78369-0. and hence I do not find any innovative information in the study
Responses to reviewer 2, comment 1:
Thank you for the suggestions and comments.
PZ-DHA is a novel compound that was synthesized in Dr. Rupasinghe’s lab. Since its synthesis, we have investigated its anti-cancer activity using several cancer models. This manuscript is part of a more extensive project that was conducted to study the anti-breast cancer activity of PZ-DHA. The innovative information contained in the present study reveals the impact of PZ-DHA on the formation of tumors and on the viability of paclitaxel-resistant breast cancer cells. These are new findings that have not previously been reported.
- All experiments were carried out using MCF-7 and MDA-MB-231-TXL. However, an additional control cell line (MCF-10) should be used to interpret and validate the key findings of the study.
Responses to reviewer 2, comment 2:
We have already conducted the experiments and published data we obtained using MCF-10A cells showing that the effect of PZ-DHA is selective toward breast cancer cells (Fernando et al., 2016; Fernando et al., 2019).
- For western blot analysis, authors should include molecular weight and densitometry of bands (Fig 4C, 4D etc).
Responses to reviewer 2, comment 3:
Band molecular weights are now included in the manuscript. In addition, the full blots were provided to the editor and indicate the band molecular weights. We will upload them into the system.
- Authors must perform a clonogenic assay to determine the anti-proliferative properties of Phloridzin Docosahexaenoate using breast cancer cells.
- Authors did not perform cell proliferation markers (Ki67, PCNA, BrdU, etc.), and cell cycle markers using the given cell lines to confirm the anti-proliferative properties of Phloridzin Docosahexaenoate.
Responses to reviewer 2, comments 4&5:
The anti-proliferative effects of PZ-DHA have already been shown and published. In brief, we have shown that PZ-DHA reduces the proliferation of MDA-MB-231 triple-negative breast cancer cells and also reduces the expression of the proliferation marker Ki-67 in MDA-MB-231 cell cultures (Fernando et al., 2016). Furthermore, we have already shown that PZ-DHA reduced the expression of Ki-67 both in MDA-MB-231 xenografts grown in immunocompromised mice and 4T1 tumors grown in immunocompetent mice (Fernando et al., 2019).
- The authors did not perform downstream or upstream pathways to confirm the ability of Phloridzin Docosahexaenoate to proliferate, migrate and invade the tumor.
Responses to reviewer 2, comment 6:
We have investigated the effects of PZ-DHA on multiple cell signaling pathways that affect breast cancer cell proliferation, migration and metastasis. We have previously shown that PZ-DHA inhibits Akt, MAPK, small molecular Rho GTPase, and EMT markers of triple-negative breast cancer cells (Fernando et al., 2019).
- Authors did not perform any in vivo experiments in the present study. They did not get consistent outcomes in the anti-proliferative properties of Phloridzin Docosahexaenoate in tumor-bearing breast cancer animals.
Responses to reviewer 2, comment 7:
In our previous work, we have shown that PZ-DHA inhibits the growth of orthotopically implanted MDA-MB-231 xenografts and 4T1 tumors in immunocompromised and immunocompetent mice respectively (Fernando et al., 2019).
Round 2
Reviewer 1 Report
The manuscript can be accept for publication in its current form.
Reviewer 2 Report
Accept in present form
Minor editing of English language required